# Self-reported gastrointestinal adverse effects of non-steroidal anti-inflammatory drugs in female students with dysmenorrhoea at Makerere University: prevalence, discontinuation and associated factors. a cross sectional study

Solomon Gobba ,[1] Winnie Kibone ,[1] Ronald Kiguba [2]

¹Makerere University, Kampala, Uganda
²Pharmacology and Therapeutics, Makerere University College of Health Sciences, Kampala, Uganda

**Correspondence to**
Solomon Gobba;
solomongobba@gmail.com

## ABSTRACT

**Background**  Primary dysmenorrhoea occurs in up to 50% of menstruating females. Non-steroidal anti-inflammatory drugs (NSAIDs) are the most commonly used therapeutic remedies for dysmenorrhoea in Uganda. However, NSAIDs are associated with a 3–5 fold increase in the risk of gastrointestinal (GI) adverse drug effects.

**Objectives**  We aimed to determine the prevalence and associated factors of self-reported NSAID-related GI adverse effects in female students who use NSAIDs in managing dysmenorrhoea-associated pain at Makerere University.

**Design**  A cross-sectional study.

**Setting**  Makerere University's main campus, situated North of Kampala, Uganda.

**Participants**  314 female students pursuing an undergraduate programme at Makerere University and residing in different halls of residence and hostels.

**Outcomes**  Social demographic data, menstrual history and treatment data.

**Results**  Overall, 314 valid responses were received from female students with a median age of 22 years (IQR: 18–29 years). The median age at menarche was 13 years (IQR: 9–18 years). 41% (n=129/314) of the respondents had used medication for dysmenorrhoea and 32% (n=41/129) of whom reported NSAID-associated GI adverse effects with nausea being the most frequently reported (44%, n=18/41)

Factors independently associated with GI adverse effects were: age at menarche (p=0.026), duration of menstruation (p=0.030) and use of ibuprofen (p=0.005). Females taking ibuprofen for dysmenorrhoea were about four times as likely to have NSAID-associated GI adverse effects (adjusted OR 3.87, 95% CI 1.51 to 9.91) than those who did not receive ibuprofen. Logistic regression was used to determine factors associated with self-reported adverse effects of NSAIDs among the female students. A p<0.05 was considered statistically significant.

**Conclusion**  We found a considerably high prevalence of NSAID-related GI adverse effects driven by factors such as age at menarche and ibuprofen use.

## STRENGTHS AND LIMITATIONS OF THIS STUDY

⇒ The study focuses on a public health issue (non-steroidal anti-inflammatory drugs (NSAIDS)-related gastrointestinal (GI) effects) in a specific population (female students with dysmenorrhoea) in Uganda, where this information is limited.

⇒ The study has clear objectives, elaborate study design and a well-defined methodology to determine the prevalence and associated factors of NSAID-related GI effects.

⇒ The online survey method excludes students without internet access, potentially leading to a biased sample that may not represent the entire student population.

⇒ Relying on self-reported data on medication use and GI effects can introduce some bias due to potential recall errors or misinterpretations.

## INTRODUCTION

Dysmenorrhoea is defined as painful menstrual cramps and is classified into primary dysmenorrhoea without any evident pathology and secondary dysmenorrhoea with a pathology.[1 2] Pinpointing the exact prevalence remains elusive, as studies conducted in diverse populations show varying rates, spanning from 20% to 94%[3] Primary dysmenorrhoea occurs in up to 50% of menstruating females and causes significant disruption in quality of life and absenteeism from school and workplaces affecting their productivity.[4] The pain is attributed to uterine prostaglandins released by endometrial cells during uterine wall sloughing before the onset of menstruation.[5] On account of the high prevalence of dysmenorrhoea with a global prevalence ranging between 20% and 90%, dysmenorrhoea is a serious public health burden.[6] Studies have documented an equally

**Table 1** Demographic and menstrual characteristics of participants

| Variable | Frequency (n) | % |
|---|---|---|
| Age of respondents, years (median, IQR) | 22 | 21–23 |
| 18–21 | 70 | 22.3 |
| 22–25 | 168 | 53.5 |
| 26–29 | 40 | 12.7 |
| 30 and above | 36 | 11.5 |
| Age at menarche, years (median, IQR) | 13 | 12–14 |
| 9–11 | 24 | 7.6 |
| 12–14 | 206 | 65.6 |
| 15 and above | 84 | 26.8 |
| Type of family | | |
| Nuclear | 194 | 61.8 |
| Extended | 81 | 25.8 |
| Single parent | 37 | 11.8 |
| Divorced parents | 2 | 0.6 |
| History of pregnancy | | |
| Yes | 26 | 8.3 |
| No | 288 | 91.7 |
| Awareness on dysmenorrhoea | | |
| Yes | 310 | 98.7 |
| No | 4 | 1.3 |
| Regular menstrual periods | | |
| Every time | 173 | 55.1 |
| Most times | 91 | 29 |
| Sometimes | 34 | 10.8 |
| Rarely | 16 | 5.1 |
| Duration of menstrual periods (days) | | |
| 1–2 | 10 | 3.2 |
| 3–4 | 95 | 30.3 |
| 5–6 | 76 | 24.2 |
| 7–8 | 19 | 6.1 |
| 9 and above | 1 | 0.3 |

higher prevalence of dysmenorrhoea estimated at 65.4%, 72.7% and 83.6% in Egypt, Ghana and Turkey, respectively.[7–9] In Uganda, the prevalence of dysmenorrhoea has been estimated at 75.8%.[10]

Non-steroidal anti-inflammatory drugs (NSAIDs) remain the most commonly used therapeutic remedies for dysmenorrhoea in Uganda because of their affordability, convenience of use and efficacy.[11] NSAIDs are commonly used as over-the-counter drugs and may often not be prescribed.[12 13] NSAIDs relieve symptoms in up to 70% of women when used correctly. Furthermore, NSAIDs are the recommended first-line treatment in management of dysmenorrhoea unless there are contraindications such

as history of hypersensitivity to aspirin or other NSAIDs, serious comorbidity and gastrointestinal (GI) ulcers or bleeding.[5]

The utilisation of NSAIDs has been linked to a significant 3–5 fold increase in the risk of GI adverse effects.[14] Dyspepsia with pyrosis, abdominal pain, nausea, anorexia, gastric erosions, ulcers, perforation, GI haemorrhage, which may result in anaemia, are some of the commonly observed GI adverse effects of NSAID therapy.[15] Studies have explored GI complications of NSAID-related treatment in cases of rheumatoid arthritis and osteoarthritis in the elderly[16] but little is known in relation to GI adverse drug effects related to NSAIDs therapy in the management of dysmenorrhoea. Furthermore, university students practice more self-medication using NSAIDs for dysmenorrhoea. In fact, a recent publication showed a high prevalence of self-medication estimated at 37% commonly mefenamic acid (58%).

This study aimed to determine the commonly used NSAIDs, the prevalence of self-reported NSAID-related GI adverse effects and the factors associated with self-reported NSAID-related GI adverse effects during the management of dysmenorrhoea among female students at Makerere University, Kampala, Uganda.

## METHODS

### Study design
A descriptive cross-sectional study was conducted, between 14 October 2020 and 29 October 2020 to determine the prevalence and factors associated with GI adverse effects to NSAIDs usage in the management of dysmenorrhoea among female students at Makerere University.

### Study setting
The study was conducted at Makerere University's main campus, situated North of Kampala, the capital city of Uganda, at the coordinates 0.3293° N, 32.5711° E. Makerere University is the largest public university in Uganda, attracting students from various regions of the country. The university accommodates over 40000 students, both resident and non-resident. The main campus features twelve Halls of Residence, with six for male students and three for female students. Medical students are accommodated at the Mulago Hospital Complex. Approximately 5000 resident undergraduates and 100 graduate students, accounting for 13% of the university's registered students, reside in the Halls of Residence. The remaining students either live in private hostels or commute from their homes. The age range of the majority of students falls between 20 and 30 years, encompassing a diverse representation of socioeconomic statuses.

### Study population
Female students pursuing an undergraduate programme at Makerere University and residing in different halls of residence and hostels. The sample size of 314 participants

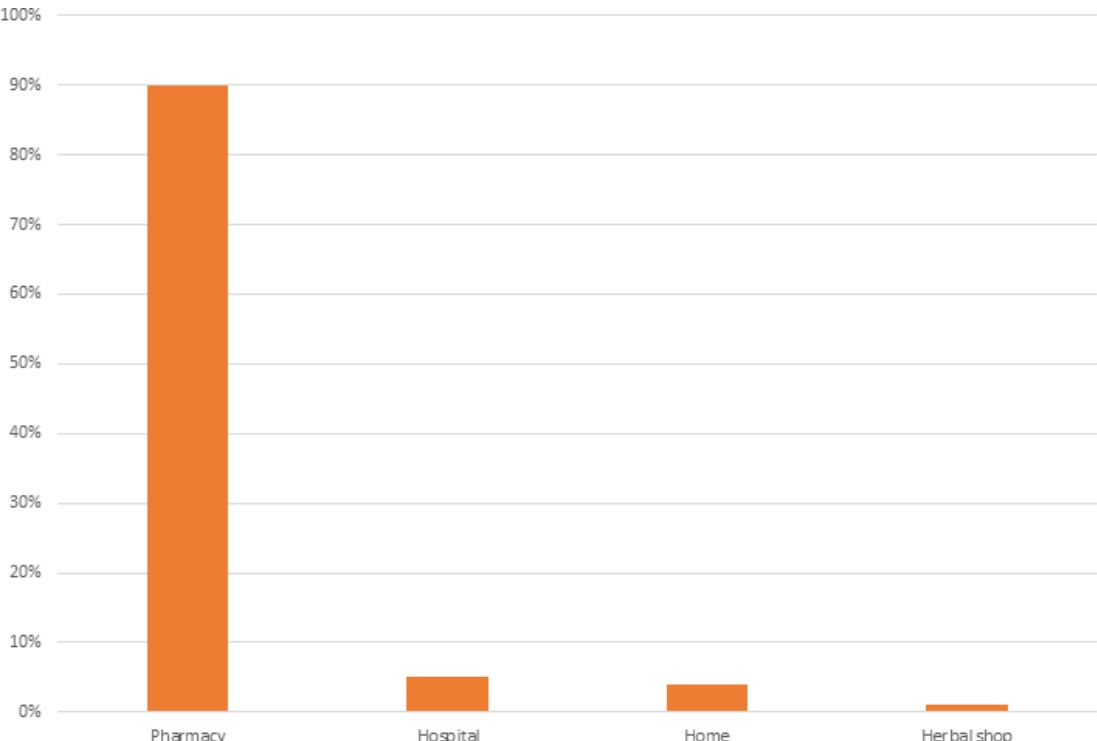

**Figure 1** Sources of medication for menstrual cramps relief among female students at Makerere University.

was calculated using the modified Kish and Leslie formula[17] for finite population size, with an estimated prevalence of dysmenorrhoea among university students (75.8%), and a 10% adjustment was made to cater for non-response.

### Selection criteria

All registered female students of Makerere University aged 18 years and older were eligible for the study after informed consent. A purposive sampling technique was used to select students for the study. Female students with primary or secondary amenorrhoea, in menopause, with mental incapacitation or without access to internet were excluded from the study.

### Data collection

We used a self-administered questionnaire to collect data for this study. The questionnaire was pretested for comprehension and appropriateness among 15 students and adjusted accordingly to ensure quality data collection. The data were collected online via a Google survey document where participants received the survey information via a link shared with them in their WhatsApp groups and in their personal email inboxes. Eligible participants consented to the study and completed the survey. The submitted survey responses were automatically stored in an online data bank, accessible only to the principal investigator (PI) and the supervisor. Additionally, the data were backed up on the PI's Google Drive to ensure data security. The questionnaire variables were constituted in three parts: sociodemographic information, menstrual history and treatment variables. These

sections aimed to gather comprehensive data relevant to the study objectives.

### Study variables

Independent variables: social demographics (age, and history of pregnancy), menstrual history (dysmenorrhoea, menarche, regularity and duration of menstrual periods) and treatment (painkillers used and their source, side effects).

### Dependent variables: NSAID-related GI adverse effects

#### Data quality control

To ensure that high-quality data were captured from the volunteers, the data collection tool was pretested and adjusted to ensure convenience and ease of manoeuvre. Checks were incorporated within the document to ensure all questions were responded to and correctly answered to reduce the margins of error.

#### Data management

Data collection tools were checked for completeness and instantly submitted to a central data bank that tallied all responses received from the participants. The central databank was synchronised and backed up on the PI's Google Drive for secondary storage. On completion of the survey, the data were translated into Excel for data cleaning and initial analysis. The data were then exported to STATA Software (StataCorp) for further analysis.

#### Data analysis

All analyses were performed using Microsoft Excel 2016 and STATA Software. Numerical data were summarised as

**Table 2** Patterns of dysmenorrhoea and medication use among female students at Makerere University

| Variable | Frequency | % |
|---|---|---|
| **Experienced menstrual pains/cramps/dysmenorrhoea** | | |
| Every time | 132 | 42.0 |
| Most times | 71 | 22.6 |
| Sometimes | 76 | 24.2 |
| Rarely | 35 | 11.2 |
| **Frequency of use of medicines (pain killers) to reduce pain during your menstrual period** | | |
| Every time | 71 | 22.6 |
| Most times | 52 | 16.6 |
| Sometimes | 60 | 19.1 |
| Rarely | 131 | 41.7 |
| **Experienced menstrual cramps in the past 3 months** | | |
| Yes | 273 | 86.9 |
| No | 41 | 13.1 |
| **How long ago the most recent period was (days)** | | |
| 1–5 | 180 | 57.32 |
| 6–10 | 98 | 31.21 |
| 11 and above | 36 | 11.47 |
| **Use of medication** | | |
| Yes | 129 | 47.3 |
| No | 144 | 52.8 |
| **Medicines used for menstrual pains/cramps/dysmenorrhoea** | | |
| Ibuprofen | 52 | 40.3 |
| Paracetamol | 52 | 40.3 |
| Diclofenac | 38 | 29.5 |
| Piroxicam | 26 | 20.2 |
| Mefenamic acid | 9 | 7.0 |
| Aspirin | 6 | 4.7 |
| Indomethacin | 3 | 2.3 |
| Dynapar | 3 | 2.3 |
| Brustan | 3 | 2.3 |
| Tramadol | 2 | 1.6 |
| Herb | 1 | 0.8 |
| Meloxicam | 1 | 0.8 |

means and SD if normally distributed; otherwise, median and IQRs were used for non-normally distributed data. Categorical data were summarised as frequencies and proportions with their 95% CIs. Associations between independent variables and dependent variables were assessed using the $\chi^2$ test and multivariable analysis using STATA V.15.1 software. A $p<0.05$ was considered statistically significant.

## Patient and public involvement

Patients and the public were at the centre of this research, our research questions and outcome measures were shaped by findings from previous studies which often reflect patient-reported outcomes, adverse events and medication-related errors. We aimed to develop research objectives that were meaningful, relevant and actionable in addressing medication safety issues from a patient-centred standpoint.

We conducted a questionnaire pretest session with a select group of participants to refine study questionnaire which ensured that the study design was sensitive to patient preferences and experiences.

The results of the study will be disseminated to study participants and the general public through various channels to ensure accessibility and comprehension. This will include the distribution of lay summaries written in plain language, presentations at patient-focused conferences and publication of study findings in open-access journals.

## RESULTS

### Characteristics of participants

Overall, a total of 314 participants responded to the study. The median age was 22 years (IQR: 21–23). The majority came from nuclear families (62%, 194/314). Only 8% (26/314) of the participants had a history of pregnancy. The median age at menarche was 13 years (IQR: 12–14) with a range of 9–18 years and more than half (55%, 173/314) experienced regular periods every time. Nearly all female students were aware of dysmenorrhoea (99%, 310/314). Table 1 summarises the baseline characteristics of participants.

### Patterns of dysmenorrhoea and medication use

The majority of participants (42%, 132/314) reported experiencing menstrual cramps every time and about a quarter (23%, 71/314) reported using medicines to reduce pain during menstrual pain every time. Most (87%, 273/314) of the participants reported experiencing menstrual cramps in the past 3 months of whom nearly half (47%, 129/273) reported using a medication for menstrual cramps with Ibuprofen and paracetamol being the most frequently used medications (40%, 52/129 and 40%, 52/129, respectively). The majority (90%, 116/129) bought the medication from pharmacies. Figure 1 shows the sources of the medication. Table 2 summarises the patterns of dysmenorrhoea and medication use among female students at Makerere University.

### Prevalence of NSAID-related GI adverse effects

Data from 129 female students who had used a medication while having dysmenorrhoea in the past 3 months were analysed. The prevalence of NSAID-associated GI adverse effects was 32% (n=41/129). Nausea (44%, n=18/41), ulcers (39%, n=16/41) and diarrhoea (39%, n=16/41) were the most frequently reported adverse, effects (figure 2). Ibuprofen (59%, n=24/41), diclofenac (46%, n=19/41), paracetamol (15%, n=6/41), indomethacin (7%, n=3/41) and aspirin (2%, n=1/41) were implicated for the adverse effects.

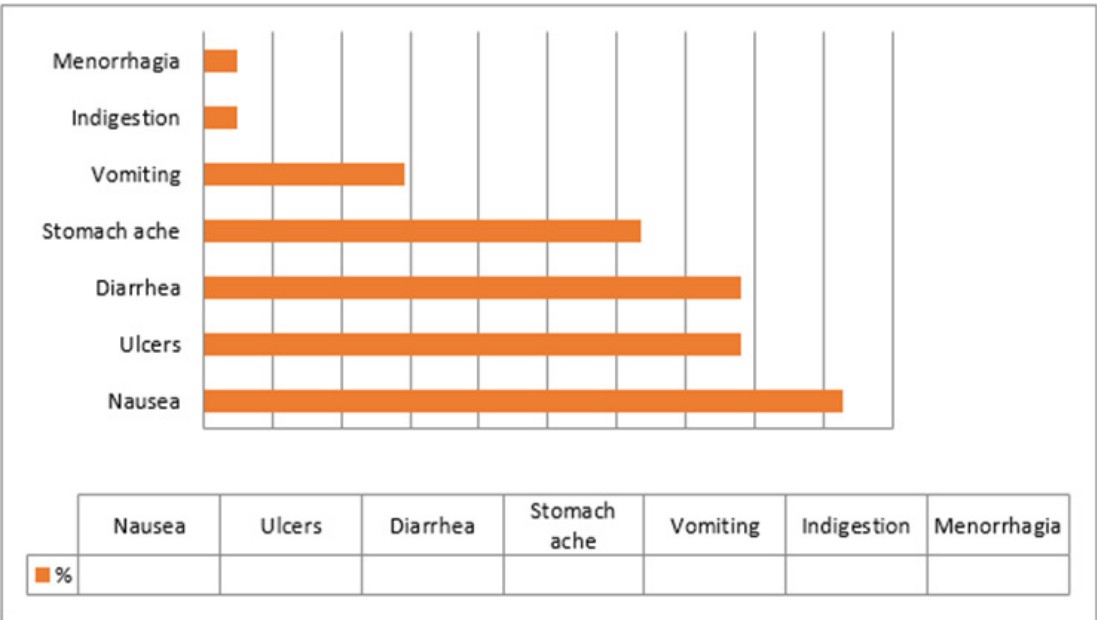

**Figure 2** Frequency of NSAID-associated GI adverse effects among female students at Makerere University. GI, gastrointestinal; NSAID, non-steroidal anti-inflammatory drug.

### Factors associated with NSAID-associated GI adverse effects

At multivariable logistic regression (table 3), factors independently associated with GI adverse effects were as follows: age at menarche (adjusted OR, AOR 1.57, 95% CI 1.05 to 2.34), the duration of menstruation (AOR 1.57, 95% CI 1.05 to 2.36) and use of ibuprofen (AOR 3.87, 95% CI 1.51 to 9.91). Females taking ibuprofen for dysmenorrhoea were about four times more likely to have NSAID-associated GIT adverse effects (AOR 3.87, 95% CI 1.51 to 9.91).

### Discontinuation for NSAID-associated GI adverse effects

Overall, 68% (88/314) of the female students reported having been advised to discontinue NSAIDs (table 4). Ulcers were the most common reason for discontinuation in 29 students (33%, 29/88). Of concern, some (8%, 7/88) of the students claimed that NSAIDs could cause infertility.

**Table 3** Multivariable logistic regression showing factors associated with NSAID-associated GI adverse effects among female students at Makerere University

| Variable | AOR | 95% CI | P value |
|---|---|---|---|
| Current age | 1.00 | 0.74 to 1.34 | 0.994 |
| Age at menarche | 1.57 | 1.05 to 2.34 | 0.026 |
| Duration of menstruation | 1.57 | 1.05 to 2.36 | 0.030 |
| Regular menstrual periods | | | |
| Every time | 1.00 | | |
| Most times | 1.33 | 0.49 to 3.59 | 0.574 |
| Sometimes | 0.28 | 0.03 to 2.76 | 0.275 |
| Rarely | 5.87 | 0.37 to 93.35 | 0.210 |
| Use of NSAID for pain relief | | | |
| Every time | 1.00 | | |
| Most times | 0.69 | 0.25 to 1.90 | 0.470 |
| Sometimes | 0.37 | 0.09 to 1.61 | 0.187 |
| Ibuprofen | 3.87 | 1.51 to 9.91 | 0.005 |

AOR, adjusted OR; GI, gastrointestinal; NSAID, non steroidal anti-inflammatory disease.

**Table 4** Discontinuation of NSAID usage among female students at Makerere University

| Variable | Frequency | % |
|---|---|---|
| Received advice to stop using a medication because of possible GI adverse effect | | |
| Yes | 88 | 68.0 |
| No | 41 | 32.0 |
| Reason for advice to stop use of the NSAID? | | |
| Ulcers | 29 | 33.0 |
| Infertility | 7 | 8.0 |
| Tolerance | 3 | 3.4 |
| Not good for women | 2 | 2.3 |
| Allergy | 1 | 1.1 |
| Can cause fibroids | 1 | 1.1 |
| Can cause cancer | 1 | 1.1 |
| Headache | 1 | 1.1 |
| Nausea | 1 | 1.1 |
| Menorrhagia | 1 | 1.1 |
| Vomiting | 1 | 1.1 |
| Nephrotic syndrome | 1 | 1.1 |

GI, gastrointestinal; NSAID, non-steroidal anti-inflammatory drug.

## DISCUSSION

This study aimed to determine the prevalence and factors associated with self-reported NSAIDs-related GI adverse effects among female students with dysmenorrhoea-associated pain at Makerere University. We found a high prevalence of GI adverse effects, reaching 32%. The factors significantly associated with adverse effects included age at menarche, duration of menstruation and ibuprofen use for pain relief.

Whereas there is a paucity of data on NSAID-related GI adverse effects among female university students, previous studies conducted in India, Pakistan and Japan among patients using NSAIDs reported a prevalence of GI adverse effects, such as peptic and gastric ulcers, estimated at 30.08%, 14.7% and 16.7%, respectively.[18–20] In contrast to our findings, previous studies have shown that diclofenac was the most common NSAID associated with GI adverse effects.[18] The difference could be attributed to variations in the most commonly used NSAID among participants in different studies and variations in study designs. A network meta-analysis by Feng and Wang revealed that tiaprofenic acid and mefenamic acid were indicated as the safest NSAIDs drugs, whereas indomethacin was the least safe with a higher likelihood to cause mild GI discomfort.[21] In addition, a significant increase of about 2.5%–5.0% in the occurrence of GI adverse effects has been observed in individuals with a history of GI incidents.[22]

Comparable to findings from Ghana where the prevalence of medication use for dysmenorrhoea was 46%, our study observed that 47% (129/273) of participants used medications.[8] However, the frequency of the use of medication to alleviate pain greatly varied among the participants and could be attributed to the intensity of pain and unregulated drug use as indicated by our finding that about 90% of participants acquired the painkillers over the counter at the pharmacy.[23] Contrary to a study by Moore *et al* showed that ibuprofen was a much safer drug than paracetamol, our study showed that females taking ibuprofen for dysmenorrhoea were about four times more likely to have NSAID-associated GIT adverse effects (59%) than paracetamol (15%).[24]

Whereas our study deduced that 1-year increase in the age at menarche and 1-day increase in the duration of menstruation were 1.57 more likely associated with having NSAID-associated GIT adverse effects, other factors implicated in other studies include *Helicobacter pylori* infection, high-dose or multiple NSAID use, history of upper GI injury, receiving hemodialysis and anticoagulant, oral corticosteroid or selective serotonin reuptake inhibitor use.[25–28]

GI adverse effects are associated with increased physician visits, risk of GI surgery, high healthcare costs and poorer health-related quality of life.[29] However, potential strategies such as the use of GI sparing NSAIDs, use of mucoprotective drugs and alternative medications or no pharmacological remedies for pain relief have been adopted in order to prevent NSAID-induced GI adverse effects.[30 31]

Our study contributes to the existing literature on the current status of NSAID-related GI effects among female university students in Uganda. However, there were some limitations. Due to the COVID-19 pandemic, the study was conducted online. Therefore, a limited number of students with internet access participated. This also limited the nature of study participant selection as we were unable to use probability sampling methods, the sampling technique used may not have given us the best representative study population. In addition, the exclusive use of a quantitative design in our study prevented us from exploring important perspectives that could have been derived from a qualitative study. While we performed statistical analyses to examine variable associations, we did not explicitly test for potential confounding and interaction effects, which may exist and have the potential to influence our findings.

## CONCLUSION

In this study, we found a considerably high prevalence of NSAID-related GI adverse effects associated with; age at menarche, longer duration of menstruation and ibuprofen use among female students at Makerere University who experienced dysmenorrhoea. Therefore, efforts directed towards mass education of females about the prevention, early identification and management of NSAID-related GI adverse effects are necessary. We recommend further exploration of alternative strategies for pain relief for females with dysmenorrhoea.

**Acknowledgements** The authors acknowledge the contribution of the following team of individuals in the data collection process. The team is as follows; Dr Haruna Kiryowa, Mr David Bakka, Mr Gerald Akena, Mr Samuel Ssenkungu, Ms Catherine Aiya Lalam and Ms Gillian Sheila Amuge.

**Contributors** All authors contributed equally. SG, WK and RK contributed to the study concept and design and developed the questionnaire. SG and WK recruited collaborators for survey distribution and data collection. SG, RK and WK contributed to data acquisition, analysis and interpretation. WK wrote the initial draft and all authors participated in subsequent writing and critical revision of the manuscript. RK oversaw the project implementation. WK managed project administration. SG was the guarantor. All authors approved the final version to be published and are accountable for all aspects of the work.

**Funding** This research was financially supported by Medical Research Council (MR/V03510X/1).

**Competing interests** None declared.

**Patient and public involvement** Patients and/or the public were not involved in the design, or conduct, or reporting, or dissemination plans of this research.

**Patient consent for publication** Not applicable.

**Ethics approval** This study involves human participants and was approved by School of Biomedical Sciences Research and Ethics Committee, Makerere University, REC approval number SBS-REC-888. Participants gave informed consent to participate in the study before taking part.

**Provenance and peer review** Not commissioned; externally peer reviewed.

**Data availability statement** Data are available on reasonable request. If you wish to reuse any or all of this article, data are available on reasonable request.

**ORCID iDs**
Solomon Gobba http://orcid.org/0009-0002-9881-6341
Winnie Kibone http://orcid.org/0000-0001-8837-1954
Ronald Kiguba http://orcid.org/0000-0002-2636-4115

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
