## [Reviewer comments · BMJ Open]

ARTICLE DETAILS

TITLE (PROVISIONAL)	Self-reported gastrointestinal adverse effects of non-steroidal anti-inflammatory drugs in female students with dysmenorrhea at Makerere University: Prevalence, discontinuation and associated factors: A cross sectional study
AUTHORS	Gobba, Solomon; Kibone, Winnie; Kiguba, Ronald

VERSION 1 – REVIEW

REVIEWER	Jarernsripornkul, Narumol Khon Kaen University
REVIEW RETURNED	20-Oct-2023

GENERAL COMMENTS	Thank you for the opportunity to review this manuscript. There are comments that the authors need to address. 1. Abstract: • Provide more data collection details in Methods• The last sentence of abstract is not related to the study findings. 2. Introduction • Previous studies about NSAIDs-induced GI adverse effects in dysmenorrhea should be mentioned.• Why did the study emphasize on the University students? 3. Methods • How were the students selected into the study? Please explain more details in sampling methods. Contents at Line 44-45 should be brief and some points are not necessary.• How to distribute the questionnaire to the participants?• Please explain more detail about sample size calculation. Whether the number of participants represented the population or not?• Did the study perform validity of the questionnaire before piloting?• How did the students specify the adverse effects of NSAIDs: eg. Closed or open questions or checklist? More details of the questionnaire contents are needed. 4. Results • How many questionnaires were distributed? What is the response rate?• “Aware of Dysmenorrhea” should be clarified? What is the awareness?• Prevalence of NSAID-related to adverse effects was used 129 as a denominator. Please clarify term of prevalence.• Table 3: are there other NSAIDs shown the significant AOR? Please specify.• “Ulcers” : how did the students report and specify Ulcers? The ulcer is not the symptom reported. 5. Discussion • Line 36-41 page 14: discussion of the study is not indicated only GI
--

	adverse effects but described in generally. It is not comparable to this study. Previous studies related to NSAIDs used in dysmenorrhea should be mentioned. • Limitation about the sampling methods should be added.
--	--

REVIEWER	Asafo, Akowuah Kwame Nkrumah University of Science and Technology, Department of Agricultural Economics, Agribusiness and Extension
REVIEW RETURNED	19-Dec-2023

GENERAL COMMENTS	Self-reported gastrointestinal adverse effects of non-steroidal anti-inflammatory drugs in female university students with dysmenorrhea: Prevalence, discontinuation and associated factors The title is well-defined with much precision Abstract The section is well structured and written. However, authors fail to report on global prevalence. Authors are advised to correct a few errors in this section Background It's interesting to note how authors bring to bear the common perceived need for this study as claims are made in other African countries like Ghana and Egypt. However, the problem statement is not clearly established; hence, authors are advised to clearly establish the claim for this study since a number of studies on the subject matter are found in the available literature. Secondly, authors are advised not to number the aims of the study in the last paragraph of the section Methods The section is succinct and on point and well established, however, on the study population sub-section, authors indicate a population of 310 but in the abstract, it is 314. I presume the 310 is erroneously indicated. Authors are to make necessary correction. The guidelines employed by authors in undertaking the processes are well-grounded and universally acceptable, with refined analyses make the paper a well-grounded one and as a serious academic paper. Discussion The discussion section is nicely structured; however, a few portions are poorly cited. Authors are advised to make amends. In future terms, there is the need for clinical trials to inform generalization of outcome regarding this interesting topic. References The section is nicely carved out as there is consistency
--

VERSION 1 – AUTHOR RESPONSE

Reviewer: 1
Dr. Narumol Jarernsiripornkul,
Khon Kaen University

Abstract

- 1 Provide more data collection details in Methods -Thank you for this comment. More details have been provided.
- 2 The last sentence of abstract is not related to the study finding -Thank you for this, we agree with

your observation. We agreed to remove this statement.

Introduction

3 Previous studies about NSAIDs-induced GI adverse effects in dysmenorrhea should be mentioned - Thank you for this comment. Prior studies have been included and cited.

4 Why did the study emphasize on the University students? - Thank you. University students are commonly prone to self-medication using NSAIDs. In addition, university students were the more easily accessible population for this study. This has been included in the manuscript

Methods

5 How were the students selected into the study? Please explain more details in sampling methods. Contents at Line 44-45 should be brief and some points are not necessary. - Purposive sampling technique was used to select students into the study. The google form document was shared to various student platforms.

6 How to distribute the questionnaire to the participants? - We shared the link to the google form to various student platforms and WhatsApp groups.

7 Please explain more detail about sample size calculation. Whether the number of participants represented the population or not? - The sample size was determined using Kish Leslie's formula. We considered a prevalence of 75.8% based on a previous study

8 Did the study perform validity of the questionnaire before piloting? - Yes, the questionnaire was pre-tested for comprehension and appropriateness among 15 students and adjusted accordingly to ensure quality data collection

9 How did the students specify the adverse effects of NSAIDs: eg. Closed or open questions or checklist? More details of the questionnaire contents are needed - A checklist generated using known adverse effects of NSAIDs was provided with an option of "Others" where the students could mention what was not part of the checklist

Results

10 How many questionnaires were distributed? What is the response rate? - Thank you for this, we agree with you that we did not include a response rate in our manuscript. Based on the nature of the data collection, we shared the google form on various student platforms making it very hard to ascertain how many students were reached.

11 "Aware of Dysmenorrhea" should be clarified? What is the awareness? - Thank you. In this study, awareness refers to participants correctly understanding what dysmenorrhea is.

12 Prevalence of NSAID-related to adverse effects was used 129 as a denominator. Please clarify term of prevalence- Thank you for this question. In this we wanted to know the number of students who developed any adverse effects after using an NSAID. Of the 314 participants, 273 experienced menstrual cramps in the last 3 months 129/273 (47.3%) of whom used medication. Considering this as our denominator, 41/129(32%) participants reported having experienced an adverse effect.

13 Table 3: are there other NSAIDs shown the significant AOR? Please specify.- No, they were not significant. Thank you.

14 "Ulcers": how did the students report and specify Ulcers? The ulcer is not the symptom reported - This was self reporting of a diagnosis of peptic ulcer disease by a medical doctor.

Discussion

15 Line 36-41 page 14: discussion of the study is not indicated only GI adverse effects but described in generally. It is not comparable to this study. Previous studies related to NSAIDs used in dysmenorrhea should be mentioned. - We agree that the discussion did not have specific a lot of studies looking at NSAIDs in dysmenorrhea and GI adverse effects. We have revised this part of the discussion

16 Limitation about the sampling methods should be added. - True, we agree with you on this. We have added this to our limitations.

Reviewer: 2

Dr. Akowuah Asafo,
Kwame Nkrumah University of Science and Technology

Title

1 Self-reported gastrointestinal adverse effects of non-steroidal anti-inflammatory drugs in female university students with dysmenorrhea: Prevalence, discontinuation and associated factors
The title is well-defined with much precision - Thank you so much for taking off time to review this document. This means a lot to us. We appreciate the comment on the title.

Abstract

2 The section is well structured and written. However, authors fail to report on global prevalence. Authors are advised to correct a few errors in this section - This is noted. This information was earlier mentioned in the introduction. We have also added a statement on it to the abstract

Background

3 It's interesting to note how authors bring to bear the common perceived need for this study as claims are made in other African countries like Ghana and Egypt. However, the problem statement is not clearly established; hence, authors are advised to clearly establish the claim for this study since a number of studies on the subject matter are found in the available literature. Secondly, authors are advised not to number the aims of the study in the last paragraph of the section - Thank you for this comment. We agree with the reviewer. There is paucity of data on the prevalence of GI adverse effects of NSAIDs among female university students who have a high prevalence of using NSAIDs for dysmenorrhea. This has been elaborated on in the manuscript. Thank you for this feedback. It has been resolved.

Methods

4 The section is succinct and on point and well established, however, on the study population subsection, authors indicate a population of 310 but in the abstract, it is 314. I presume the 310 is erroneously indicated. Authors are to make necessary correction.
The guidelines employed by authors in undertaking the processes are well-grounded and universally acceptable, with refined analyses make the paper a well-grounded one and as a serious academic paper
- Thank you for this. This has been corrected.
Thank you so much for these kind comments and guidance. We do appreciate.

Discussion

5 The discussion section is nicely structured; however, a few portions are poorly cited. Authors are advised to make amends. In future terms, there is the need for clinical trials to inform generalization of outcome regarding this interesting topic. - Thank you. These have been edited.

References

6 The section is nicely carved out as there is consistency - Thank you for this, and thank you once again for taking off time to review this. We have learnt a lot from this process.